# Vitamin D status, nutrition and growth in HIV-infected mothers and HIV-exposed infants and children in Botswana

**Alyssa M. Tindall**[1], **Joan I. Schall**[1], **Boitshepo Seme**[2], **Bakgaki Ratshaa**[2], **Michael Tolle**[3], **Maria S. Nnyepi**[4], **Loeto Mazhani**[5], **Richard M. Rutstein**[6,7], **Andrew P. Steenhoff**[2,5,7,8]*, **Virginia A. Stallings**[1,7]

1 Gastroenterology, Hepatology and Nutrition, Children's Hospital of Philadelphia, Philadelphia, PA, United States of America, 2 Botswana-UPenn Partnership, Gaborone, Botswana, 3 Botswana-Baylor Children's Clinical Centre of Excellence, Gaborone, Botswana, 4 Department of Nutrition, University of Botswana, Gaborone, Botswana, 5 Department of Pediatrics and Adolescent Health, School of Medicine, University of Botswana, Gaborone, Botswana, 6 General Pediatrics, Children's Hospital of Philadelphia, Philadelphia, PA, United States of America, 7 Department of Pediatrics, Perelman School of Medicine at the University of Pennsylvania, Philadelphia, PA, United States of America, 8 Divisions of Infectious Diseases, Children's Hospital of Philadelphia, Philadelphia, PA, United States of America

* steenhoff@email.chop.edu

**Data Availability Statement:** All relevant data are within the manuscript and its Supporting Information files.

## Abstract

### Background

Poor vitamin D status is a global health problem and common in patients with human immunodeficiency virus (HIV) in high-income countries. There is less evidence on prevalence of vitamin D deficiency and nutrition and growth in HIV-infected and -exposed children in low- and middle-income countries.

### Objectives

To determine the vitamin D status in Batswana HIV-infected mothers and their children, differences among HIV-infected mothers and between HIV-exposed and -infected infants and children, and associations between vitamin D and disease-related outcomes, nutrition, and growth.

### Methods

This was a cross-sectional study of HIV+ mothers and HIV-exposed infants and unrelated children (1–7.9 years). Serum 25-hydroxyvitamin D (25(OH)D) was measured, among other nutritional indicators, for mothers, infants and children. Vitamin D status for HIV-infected mothers and children, and an immune panel was assessed. History of HIV anti-retroviral medications and breastfeeding were obtained. Data were collected prior to universal combination antiretroviral therapy in pregnancy.

**Funding:** This research was supported by a grant from the Penn Center for AIDS Research (CFAR), an NIH-funded program (P30 AI 045008) to VAS. The funders had no role in study design, data collection and analysis, decision to publish, or preparation of the manuscript.

**Competing interests:** The authors have declared that no competing interests exist.

## Results

Mothers ($n = 36$) had a mean serum 25(OH)D of 37.2±12.4ng/mL; 11% had insufficient (<20ng/mL), 17% moderately low (20.0–29.9ng/mL) and 72% sufficient ($\geq$30ng/mL) concentrations. No infants ($n = 36$) or children ($n = 48$) were vitamin D insufficient; 22% of HIV- and no HIV+ infants had moderately low concentrations and 78% of HIV- and 100% of HIV+ infants had sufficient status, 8% of HIV- and no HIV+ children had moderately low concentrations and 92% of HIV- and 100% HIV+ children had sufficient concentrations. HIV+ children had significantly lower length/height Z scores compared to HIV- children. Length/height Z score was positively correlated with serum 25(OH)D in all children (r = 0.33, p = 0.023), with a stronger correlation in the HIV+ children (r = 0.47 p = 0.021). In mothers, serum 25(OH)D was positively associated with CD4% (r = 0.40, p = 0.016).

## Conclusions

Results showed a low prevalence of vitamin D insufficiency in Botswana. Growth was positively correlated with vitamin D status in HIV-exposed children, and HIV+ children had poorer linear growth than HIV- children.

## Introduction

Vitamin D is a fat-soluble, essential vitamin obtained primarily through ultraviolent B rays and plays a critical role in the regulation of bone metabolism, cell growth, immune and neuro-muscular function. Vitamin D status correlates with numerous communicable and non-communicable diseases that are major public health problems worldwide. Poor vitamin D status is common in patients with human immunodeficiency virus (HIV) in the United States [1], but there is not substantial evidence on prevalence of vitamin D deficiency in HIV infected and exposed populations in low- and middle-income countries [2], such as Botswana [3]. Botswana has the third highest prevalence of HIV in the world and over 15% of children are HIV-exposed [4, 5]. Determining the prevalence of vitamin D insufficiency in this population is of great importance.

Vitamin D sufficiency can be effectively achieved using vitamin D supplementation [6], however, there are limited data on the prevalence of deficiency among HIV-infected mothers and no data on exposed and infected infants and young children in Botswana. Black race is a risk factor for insufficient vitamin D [7] and HIV-infected mothers are at greater risk of vitamin D deficiency due to tenofovir- and efavirenz-containing combination antiretroviral therapy (cART), which are associated with increased risk of vitamin D deficiency [8–13]. HIV-exposed children account for more than half of 24-month mortality in Botswana, but contributing factors, such as vitamin D status, have not been determined [14]. Vitamin D stores are low at birth in all infants and vitamin D is obtained through breast milk and sunlight exposure. Vitamin D deficiency is more prevalent in children with perinatally acquired HIV infections [15]. Determining the vitamin D status of HIV-infected mothers and HIV-infected or HIV-exposed infants and children in Botswana is critical. Determining if there are any confounding variables, such as feeding practices or nutritional status, that affect vitamin D concentrations will identify areas that could be targeted for intervention.

The primary objective of this urban, Botswana-based observational study was to determine the vitamin D status in HIV-infected mothers and their children–including HIV-exposed or

HIV-infected. The secondary objectives were to examine differences in vitamin D status among HIV-infected mothers and between HIV-exposed and infected infants and children, and to explore associations between vitamin D status and disease-related outcomes, breast-feeding, and growth.

## Materials and methods

### Participants

This was an observational, age- and HIV-status stratified survey of 36 HIV-infected (HIV+) mothers and their infants (0–11.9 months), and 48 unrelated children (1–7.9 years). Mothers, infants and children were recruited from Princess Marina Hospital, Gaborone, and Bamalete Lutheran Hospital, Ramotswa, in Botswana where they were followed for medical care. Mothers and children were recruited over a 10-month period from November, 2012 through August, 2013. Mothers with HIV infection and infants and children with HIV exposure in usual state of good health were eligible and consent was obtained. Mothers, infants or children with a health condition unrelated to HIV infection likely to affect nutritional status, hospitalization, accident or emergency visit, or unscheduled sick visit to a clinic in the two weeks prior to recruitment were not eligible. The protocol was approved by the Institutional Review Boards of the Children's Hospital of Philadelphia (CHOP), University of Pennsylvania, Botswana Ministry of Health, Princess Marina Hospital, Bamalete Lutheran Hospital, and Baylor College of Medicine.

### Prevention of mother to child transmission guidelines

All HIV infected pregnant women were assessed for eligibility for triple prophylaxis/cART according to Botswana's national HIV guidelines at the time. HIV infected women were evaluated for cART eligibility based on laboratory testing (CD4 cell count) or clinical presentation (WHO clinical stage). Women with CD4 cell count less than 350 or WHO clinical stage 3 or 4 were started on cART through the national ARV program as soon as possible, regardless of the stage of pregnancy. Women attending healthcare facilities that provided triple ARV prophylaxis, who were not eligible for cART were provided with triple ARV prophylaxis starting from 14 weeks of gestation. Women who chose to breastfeed continued triple ARV prophylaxis until their infants were at least six months of age and completed weaning. Both women on cART and those that were on ARV prophylaxis were given supplementary AZT 300 mg every 3 hours during labor and delivery [16].

Similarly by 2009, >95% of infants born to HIV infected mothers were given ARV prophylaxis at birth according to national guidelines; they were given nevirapine as a single dose as soon as possible after birth (within 72 hours) plus four weeks of AZT [16].

### Sampling

A convenience sample was used. For balanced representation across infant and child ages and HIV status groups, 18 mother-infant pairs with infants aged 0–5.9 months and 18 pairs with infants aged 6–11.9 months attending routine medical visits were enrolled. Given the rate of perinatal transmission of HIV in Botswana was less than 3% at the time of data collection [16], the younger infants (aged 0–5.9 months) were categorized as HIV-. Removing the younger infants did not change the results, therefore, we included them in the dataset. The older infants, ages 6–11.9 months, were balanced by HIV status as determined by the HIV PCR result in their medical record. Forty-eight unrelated children with HIV-infected mothers were

also enrolled, 24 at ages 1–3.9 years and 24 at ages 4–7.9 years, balanced by HIV status. An attempt was made to balance for equal sex representation of infants and children.

## Clinical data collection

Following consent, participant study visits consisted of a structured questionnaire, physical measurements, and a blood draw. Clinical data, including HIV status and current medications, were extracted from medical records. HIV status was classified using the Centers for Disease Control and Prevention (CDC) clinical classifications [17, 18] and CD4 count. cART regimens were characterized as tenofovir-containing, and either protease inhibitor-based (PI) or non-nucleoside reverse transcriptase inhibitor-based (NNRTI), i.e., efavirenz or nevirapine. Information on whether infants and children were breast or bottle fed was also obtained from the medical records.

Trained research nurses completed anthropometric measurements [19] and equipment was calibrated weekly. Height and length in centimeters were measured using a stadiometer (Seca, UK) and weight in kilograms by digital standing scale (Adam Medical) for adults and wheelchair digital scale (Seca, UK) for children. Anthropometric measurements were completed in triplicate and the mean was used for analysis. Body mass index (BMI unit) was calculated as $kg/m^2$. For infants and young children up to age two years where length was measured, length and weight were converted into Z scores (standard deviation units) using the WHO growth charts (http://www.cdc.gov/growthcharts/), which is recommended by the CDC [20]. For older children the CDC recommends using the CDC algorithms to convert height and weight into Z scores [21].

## Laboratory methods

Vitamin D status biomarkers included serum 25-hydroxyvitamin D (25(OH)D), 1,25-dihydroxyvitamin D (1,25(OH)D), intact parathyroid hormone (PTH), and other nutritional biomarkers associated with vitamin D (serum magnesium, phosphorous, albumin, calcium). Immune status was determined in all women and those children living with HIV. Serum 25(OH)D was measured using liquid chromatography tandem mass spectrometry (Clinical Laboratory, CHOP, Philadelphia, PA, USA) with intra- and inter-assay coefficients of variation (CV) below 8%; serum 1,25(OH)D and PTH were assessed by radioimmunoassay using a radio-iodinated tracer (Heartland Assays, Ames, IA, USA) with intra- and inter-assay CV of 9.8% and 12.6% for 1,25(OH)D and 2.7% and 4.3% for PTH. Serum magnesium, phosphorus, albumin, and calcium (corrected for albumin) were measured using standard techniques (Diagnofirm Diagnostic Laboratory, Gaborone, Botswana). The immunological indicators and HIV-1 RNA VL were measured by Diagnofirm Diagnostic Laboratories (Gaborone, Botswana).

## Statistical methods

Serum 25(OH)D concentration was categorized as: insufficient, < 20 ng/mL; moderately low, 20–29.9 ng/mL; and sufficient, ≥ 30 ng/mL. CD4% was categorized as: low, < 15%; moderate, 15–24.9%; and high, ≥ 25%, where a high CD4% indicates a favorable immunological profile. Characteristics of the mothers, infants, and children were analyzed using conventional parametric statistics (means, SD, percentages) and comparisons made between HIV status groups using unpaired (two-sample) student's $t$-tests for continuous variables and chi-square or Fisher's exact tests for categorical data. Z scores for length, height and weight were compared for infants and children by analysis of covariance (ANCOVA), adjusting for chronological age, to account for growth trends with age. Breastfeeding (ever breast fed) and sex were

also examined as covariates. Season of examination was grouped into two categories: Spring/ Summer (August-January) and Autumn/Winter (February-July). Vitamin D-related laboratory values were compared using analysis of covariance (ANCOVA), adjusting for season of examination (Spring/Summer, Autumn/Winter) to account for possible differences in vitamin D related to sunlight exposure. Given the effect of efavirenz on vitamin D status [8, 9, 11–13], a unique category was created for efavirenz-containing regimens to control for confounding treatment effects. An undetectable RNA VL was set to < 25 copies/mL (RNA log = 1.4). Associations between serum 25(OH)D and outcomes were explored using Pearson correlation coefficients or Spearman rank correlations depending upon skewness. Levels of statistical significance were set at $\alpha$ = 0.05.

## Results

Thirty-six HIV+ mothers of infants aged 0–11.9 months were enrolled. HIV+ mothers gave birth to 27 (75%) HIV negative (HIV-) and 9 (25%) HIV+ infants. Forty-eight children, aged 1–7.9 years with HIV exposure (HIV+ mothers) were recruited. Half of the children recruited were HIV- and half HIV+ by study design.

The HIV-infected mothers ranged in age from 21 to 41 (mean±SD, 31.8±5.1 yrs), and had been diagnosed with HIV 3.7±3.2 yrs (range 0.5 to 10 years) prior: 69% were taking cART and 42% were on efavirenz-containing regimens (Table 1). Serum 25(OH)D was 37.2±12.4 ng/mL; 11% had insufficient (<20 ng/mL), 17% moderately low (20.0–29.9 ng/mL) and 72% sufficient (≥ 30 ng/mL) serum 25(OH)D concentrations. The mothers of HIV- vs. HIV+ infants did not differ in terms of current age, height, weight, BMI, or current use of cART, including the use of efavirenz-containing regimens (Table 1). There was a significant difference in the season of examination between mothers with HIV- or HIV+ infants. In terms of vitamin D status, even adjusting for season of examination, the mothers of infants who were HIV+ had significantly lower concentrations of serum 25(OH)D and serum albumin and had a smaller proportion of serum 25(OH)D concentrations in the sufficient range (≥ 30 ng/mL) compared to the mothers of HIV- infants (Table 1). Further adjusting for current use of efavirenz-containing regimens did not alter the significance of these results.

Comparison of the infants who were HIV- or HIV+ indicated a tendency for HIV+ infants to be smaller in both length and weight by approximately -0.9 Z score, but the differences were not statistically significant (Table 2). There were no differences in sex, and few of either group were ever breastfed. Adjusting for season of examination, there was no difference in concentration of serum 25(OH)D, and no infant, whether HIV- or HIV+ had insufficient concentration of serum 25(OH)D. Serum 1,25(OH)D was significantly higher in infants who were HIV +.

Children aged 1–7.9 years had poor growth status overall (approximately -1.0 Z scores for length/height and weight). HIV+ children were significantly shorter, after controlling for age. When examined separately, males and females had similar growth outcomes, with HIV+ children shorter than their HIV- counterparts. There were no HIV- children who were ever breastfed, but a little over half of the HIV+ were breastfed as infants (Table 2). Both sex and whether ever breastfed were tested as covariates, and neither significantly contributed to model or explained the difference in length/height z scores between HIV+ and HIV- children. Furthermore, in the HIV+ group, there were no significant differences in growth status in length or weight between children who were breastfed as infants and those that were not. Vitamin D status in HIV+ was comparable to children who were HIV-, and no child had an insufficient (< 20 ng/mL) 25(OH)D concentration. Serum calcium was significantly higher in children who were HIV+. Length/height Z score was positively correlated with serum 25(OH)

Table 1. Maternal characteristics of HIV positive mothers (n = 36) stratified by HIV status of their infants.

| Maternal Characteristic | HIV+ Mothers | | p-value |
| | HIV- Infants | HIV+ Infants | |
| | n = 27 | n = 9 | |
|---|---|---|---|
| Age, y | 32.6±5.0 | 30.9±5.0 | 0.38 |
| Height, cm | 159.6±6.3 | 158.9±7.7 | 0.78 |
| Body weight, kg | 60.6±9.8 | 57.2±9.3 | 0.36 |
| BMI, kg/m$^2$ | 23.8±3.8 | 22.6±4.0 | 0.43 |
| < 18.5, % | 4 | 11 | 0.71 |
| 18.5–24.9, % | 63 | 67 | |
| 25.0–29.9, % | 26 | 22 | |
| ≥ 30.0, % | 7 | 0 | |
| Season of examination | | | |
| Spring/Summer, % | 11 | 78 | <0.001 |
| Fall/Winter, % | 89 | 22 | |
| cART, % | 70 | 67 | 0.84 |
| Efavirenz-containing regimens, % | 37 | 56 | 0.33 |
| Serum vitamin D and nutrition indicators* | | | |
| 25(OH)D, ng/mL | 40.5±2.5 | 27.2±4.9 | 0.03 |
| < 20.0, % | 11 | 11 | 0.002 |
| 20–29.9, % | 4 | 56 | |
| ≥ 30.0, % | 85 | 33 | |
| 1,25(OH)D, pg/mL† | 41.8±4.9 | 44.8±9.4 | 0.79 |
| PTH, pg/mL† | 35.7±2.7 | 24.6±5.2 | 0.09 |
| Magnesium, mmol/L | 0.96±0.02 | 0.87±0.04 | 0.06 |
| Phosphorus, mmol/L | 1.10±0.03 | 1.14±0.05 | 0.53 |
| Calcium, mmol/L | 2.26±0.02 | 2.25±0.04 | 0.49 |
| Albumin, g/L | 39.1±0.7 | 35.6±1.3 | 0.04 |

Data are presented as mean ± standard deviation (SD) or a least square means (LSM) ± standard error of the mean (SEM) for continuous variables, and as percent for distribution of categorical variables. HIV = human immunodeficiency virus; BMI = body mass index; cART = combined antiretroviral therapy; 25(OH)D = 25-hydroxyvitamin D; 1,25(OH)D = 1,25-dihydroxyvitamin D; PTH = intact parathyroid hormone.

*Results are reported as LSM ± SEM from analyses of covariance, adjusting for season of examination.

†Sample sizes are $n = 26$ for mothers of HIV- infants and $n = 9$ for mothers of HIV+ infants.

D in all 1–7.9 year old children (r = 0.33, p = 0.023), with a stronger correlation in the HIV + children (r = 0.47 p = 0.021).

HIV+ participants were evaluated for immunological markers, with comparisons made between HIV+ mothers with HIV- and HIV+ infants (Table 3). HIV+ mothers of HIV+ infants had significantly lower CD4%, greater percentage with CD4% < 15%, and higher RNA VL. In the HIV-infected mothers as a group, serum 25(OH)D was positively correlated with CD4% (r = 0.40, p = 0.016). Table 3 also presents the HIV disease status for the 9 infants and the 24 children in this survey who were HIV+. Nearly all of the HIV+ infants (89%) and all of the HIV+ children were on cART, and half of the infants and 79% of the children had CD4% at or greater than 25%.

## Discussion

The aims of this cross-sectional, observational study were to determine the vitamin D status of HIV-infected Batswana mothers and their HIV-exposed infants and young children. Overall,

**Table 2. Infant (n = 36) and child (n = 48) characteristics stratified by HIV status.**

| | Infants | | | Children | | |
|---|---|---|---|---|---|---|
| | HIV- | HIV+ | p-value | HIV- | HIV+ | p-value |
| **Characteristic** | **n = 27** | **n = 9** | | **n = 24** | **n = 24** | |
| Age, mo | 5.1±3.6 | 8.5±2.4 | 0.02 | 42.6±15.0 | 54.3±26.0 | 0.06 |
| Length/height Z Score* | -0.89±0.37 | -1.73±0.67 | 0.31 | -0.83±0.20 | -1.43±0.20 | 0.04 |
| Weight Z Score* | -0.19±0.24 | -1.08±0.42 | 0.10 | -1.15±0.24 | -0.90±0.24 | 0.49 |
| Sex, male, % | 37 | 33 | 0.84 | 50 | 38 | 0.38 |
| Breastfed, % | 7 | 11 | 0.73 | 0 | 54 | <0.001 |
| Season of examination | | | | | | |
| Spring/Summer, % | 11 | 78 [‡] | <0.001 | 50 | 54 | 0.77 |
| Fall/Winter, % | 89 | 22 | | 50 | 46 | |
| Serum vitamin D and nutrition indicators* | | | | | | |
| 25(OH)D, ng/mL | 45.4±2.8 | 50.2±5.6 | 0.49 | 47.9±2.8 | 51.4±2.8 | 0.37 |
| < 20.0 ng/mL, % | 0 | 0 | 0.12 | 0 | 0 | 0.15 |
| 20–29.9 ng/mL, % | 22 | 0 | | 8 | 0 | |
| ≥ 30.0 ng/mL, % | 78 | 100 | | 92 | 100 | |
| 1,25(OH)D, pg/mL† | 43.6±5.4 | 82.6±12.6 | 0.02 | 51.1±5.9 | 57.2±6.1 | 0.51 |
| PTH, pg/mL† | 24.4±2.6 | 24.2±5.4 | 0.97 | 22.0±1.7 | 26.3±1.8 | 0.09 |
| Magnesium, mmol/L | 1.04±0.03 | 0.94±0.05 | 0.16 | 1.14±0.04 | 1.07±0.04 | 0.14 |
| Phosphorus, mmol/L | 1.97±0.06 | 1.94±0.05 | 0.85 | 1.59±0.04 | 1.52±0.04 | 0.24 |
| Calcium, mmol/L | 2.52±0.02 | 2.51±0.04 | 0.80 | 2.29±0.02 | 2.33±0.02 | 0.04 |
| Albumin, g/L | 37.4±0.7 | 38.1±1.3 | 0.67 | 38.0±0.5 | 38.5±0.5 | 0.49 |

Data are presented as mean ± standard deviation (SD) or a least square means (LSM) ± standard error of the mean (SEM) for continuous variables, and as percent for distribution of categorical variables. HIV = human immunodeficiency virus; 25(OH)D = 25-hydroxyvitamin D; 1,25(OH)D = 1,25-dihydroxyvitamin D; PTH = intact parathyroid hormone.

*Results are reported as least square means (LSM) ± standard error of the means (SEM) from analyses of covariance, adjusting for age (in months) for length/height z-scores and weight and for season of examination for the serum vitamin D and nutrition indicators.

†Sample sizes are n = 27 for HIV- infants and n = 8 for HIV+ infants; sample sizes are n = 24 for HIV- children and n = 23 for HIV+ children.

11% of HIV+ mothers had an insufficient vitamin D status and mothers with HIV+ infants had lower 25(OH)D concentrations compared to mothers with HIV- infants. The mothers of HIV+ infants also had poorer HIV disease status with lower CD4% and higher RNA viral load than their counterparts with HIV- infants. This was the first study of vitamin D status in Batswana infants and young children exposed to HIV and we found no infants or children were vitamin D insufficient based upon a serum 25(OH)D concentration <20.0 ng/mL, whether they were HIV+ or HIV-. Vitamin D status positively correlated with length/height Z score in both HIV+ and HIV- children and also with CD4% in HIV+ mothers.

This study contributes to understanding the prevalence of vitamin D deficiency among HIV-infected individuals in Botswana. Previous reports on vitamin D status in Batswana adults and children reported a large range of insufficiency; 17–78% of participants had insufficient vitamin D status [6, 22, 23]. Findings from the present study indicate a lower prevalence of vitamin D insufficiency in Batswana HIV+ mothers (11%) compared to previous reports in adults and children with and without HIV infections. Our results show better vitamin D status in Batswana mothers, infants and children compared to other African countries, such as Ethiopia and Algeria [24]. These differences may be due to demographics and dietary habits. One study reported consumption of traditional and indigenous Batswana foods was associated with better dietary diversity and food security [25]. Traditional Batswana foods are more nutrient

**Table 3. Immunological and clinical characteristics of HIV positive mothers, infants, and children.**

| | HIV+ Mothers | | | HIV+ | |
| | HIV- Infants | HIV+ Infants | p-value | Infants | Children |
| Characteristic | n = 27 | n = 9 | | n = 9 | n = 24 |
|---|---|---|---|---|---|
| Age at diagnosis, y | 28.4±5.4 | 28.7±4.6 | 0.86 | 0.2±0.1 | 1.3±1.3 |
| Years since diagnosis, y | 4.2±3.4 | 2.2±2.1 | 0.09 | 0.5±0.3 | 3.2±2.0 |
| cART, % | 70 | 67 | 0.84 | 89 | 100 |
| CD4 count* | 489±223 | 345±239 | 0.11 | 1771±1408 | 1341±771 |
| CD4% | 28.1±7.8 | 15.9±5.2 | <0.001 | 25.2±11.0 | 35.5±10.9 |
| < 15.0, % | 7† | 56 | | 25 | 4 |
| 15.0–24.9, % | 30 | 33 | | 25 | 17 |
| ≥ 25.0, % | 63 | 11 | 0.003 | 50 | 79 |
| CD8 count* | 807±408 | 1003±618 | 0.28 | 2173±1364 | 1171±457 |
| RNA, log copies/mL* | 2.07±1.40 | 3.63±1.81 | 0.01 | 3.10±1.53 | 1.64±0.54 |

Data are presented as mean ± standard deviation (SD), and as percent for distribution of categorical variables. HIV = human immunodeficiency virus; SD = standard deviation; cART = combined antiretroviral therapy.

*For CD4 count, sample sizes are n = 27 for mothers of HIV- infants, n = 9 for mothers of HIV+ infants, n = 8 for HIV+ infants, and n = 24 for HIV+ children. For CD8 count, sample sizes are n = 27 for mothers of HIV- infants, n = 9 for mothers of HIV+ infants, n = 8 for HIV+ infants, and n = 22 for HIV+ children. For RNA log copies, sample sizes are n = 27 for mothers of HIV- infants, n = 9 for mothers of HIV+ infants, n = 4 for HIV+ infants, and n = 15 for HIV+ children.

dense compared to Western-style foods, which are energy-dense and nutrient-poor [26]. Western foods are becoming available in Botswana and could contribute to vitamin D status [26]. Future studies exploring the relation between dietary intake and vitamin D status in HIV + individuals may provide insight to the range of vitamin D insufficiency reported in Botswana and other African countries.

The proportion of vitamin D insufficiency we observed in HIV+ mothers was not reflected in HIV-exposed and infected infants in our study. No infants were deficient, regardless of whether infants were breastfed or not, and although mothers with HIV- infants had significantly higher 25(OH)D concentrations, there were no significant differences in 25(OH)D concentrations between HIV+ and HIV- infants. This was an unexpected finding given the group of mothers with insufficient vitamin D concentrations and breastfed infants, which supplies minimal vitamin D. Another study in Botswana reported 19% of infants and children under 2 years (n = 80) with and without tuberculosis were vitamin D insufficient [27] and a study in otherwise healthy Tanzanian infants reported 76% had insufficient vitamin D [28]. HIV + infants in the present study had significantly greater 1,25(OH)D compared to HIV-, however, 1,25(OH)D is not considered a good surrogate for vitamin D status [29] and so the implications of this difference are unclear. Similarly, there were no differences in vitamin D status between HIV+ or HIV- children. There was a statistically significant difference in calcium concentrations between HIV+ and HIV- children, but the difference is not clinically significant and both groups had concentrations in a safe range. Findings from this study show HIV-exposed infants and children in Botswana are not vitamin D insufficient according to the < 20 ng/mL cut-point.

Serum 25(OH)D was also correlated with length/height Z-score in HIV+ and HIV- children 1–7.9 years; children with better Z-scores had higher vitamin D concentrations. There is existing evidence that low serum 25(OH)D is more prevalent in stunted populations [30, 31], which may be related to the integral role of vitamin D in muscle and bone metabolism [32]. However, maternal vitamin D status versus child vitamin D status may differentially affect the risk of growth faltering. Sudfeld and colleagues [33] reported low vitamin D concentrations

(<10 ng/mL) in Tanzanian infants exposed to HIV were associated with wasting. However, Powis et al. [22] reported no association between maternal insufficient vitamin D concentrations (<32 ng/mL) and infant growth in Botswana. Identifying variables that affect vitamin D status in infancy and early childhood, such as dietary vitamin D, would help understand the relation between vitamin D status and stunting in HIV-exposed Batswana offspring. For example, HIV-exposed and–infected children may have generally poorer health and appetite, which could affect their intake. Findings from our survey demonstrate the potential impact of vitamin D concentrations on childhood growth. Even when vitamin D concentrations were sufficient, the correlation between vitamin D and growth status was still significant.

Our results also show a correlation between vitamin D status and HIV disease status. This finding agrees with existing literature that showed vitamin D supplementation improved HIV disease status [6, 34–36]. We found a significant, positive correlation between serum 25(OH)D and CD4% in the HIV infected mothers. This correlation suggests adequate vitamin D status may be associated with improved HIV disease status. Although there was a small percentage of mothers with insufficient vitamin D status, the immunological benefits of vitamin D may only be observed at sufficient or greater plasma 25(OH)D concentrations. Data were collected prior to universal cART in pregnancy, which may have affected the mean CD4%. The present study echoes the favorable relation between vitamin D status and immune outcomes reported elsewhere [34, 35].

There were significant differences in length/height Z-scores between HIV+ and HIV- children; HIV+ children had poorer linear growth than HIV-uninfected children aged 1–7.9. The HIV+ infants had lower length Z scores by almost a full Z score, although this did not reach significance due to small sample size. It is important to recognize there may be confounding variables, such as diet, that were not captured as part of this study. However, these results corroborate with other reports of stunting among HIV+ children [33, 37, 38]. Overall, our results show HIV-exposed infants and children had poorer overall growth regardless of whether they are HIV- or HIV+. Sudfeld et al. [33] reported HIV-exposed Batswana children had an increased risk of stunting compared to HIV-unexposed peers in children less than 5 years old. Linear growth faltering serves as a marker of malnutrition and is associated with increased morbidity and mortality [39]. Our survey is in agreement with the current literature and showed HIV+ children had poorer linear growth compared to HIV- children.

Strengths of this study include the under-studied population of infants and children exposed to HIV in Botswana and nutrition and growth-related outcomes explored in relation to both HIV and vitamin D status. The study limitations were the difference in the season of examination between the two cohorts of mothers with HIV- or HIV+ infants, though this was adjusted for during statistical analysis using ANCOVA, and there may be confounding variables that were not captured in this cross-sectional study. Additionally, we used a convenience sample recruited in 2012–2013 that may not be representative of the population, the sample size was modest and future larger studies are needed to confirm these data.

## Conclusion

In this urban Botswana-based observational study, only 11% of HIV-infected mothers exhibited insufficient serum vitamin D status, yet none of their infants were vitamin D insufficient. Further, no HIV-exposed children 1–7.9 years were vitamin D insufficient. Children living with HIV had poorer linear growth than HIV-uninfected children, with significantly lower length/height status in 1–7.9 year old children. However, additional data such as diet may provide a better understanding of HIV status on growth. There was also a significant association between vitamin D status and HIV disease status among mothers living with HIV. Results

from this study show a low prevalence of vitamin D insufficiency in a smaller cohort and emphasize the need for further research to identify confounding variables that affect vitamin D status in HIV infected and exposed individuals.

## Supporting information

**S1 Data. Botswana-vitamin D study data.**
(XLSX)

## Acknowledgments

Authors are grateful to the subjects and their families for study participation. In addition we wish to thank our colleagues in the Ministry of Health who provide day-to-day care for these patients, as well as to the leadership of all organizations involved whose work makes cross-institutional partnerships and collaborations possible, including Dr. Marape Marape who played a role in enabling some of the work with the Botswana Baylor Center of Excellence. The authors would also like to recognize Julia Samuel and Dr. Mary Hediger for their extensive help on this project.

## Author Contributions

**Conceptualization:** Joan I. Schall, Andrew P. Steenhoff, Virginia A. Stallings.

**Formal analysis:** Alyssa M. Tindall, Joan I. Schall, Andrew P. Steenhoff, Virginia A. Stallings.

**Funding acquisition:** Virginia A. Stallings.

**Investigation:** Joan I. Schall, Boitshepo Seme, Bakgaki Ratshaa, Michael Tolle, Maria S. Nnyepi, Loeto Mazhani, Richard M. Rutstein, Andrew P. Steenhoff, Virginia A. Stallings.

**Methodology:** Joan I. Schall, Andrew P. Steenhoff, Virginia A. Stallings.

**Project administration:** Joan I. Schall, Andrew P. Steenhoff, Virginia A. Stallings.

**Supervision:** Andrew P. Steenhoff, Virginia A. Stallings.

**Writing – original draft:** Alyssa M. Tindall, Joan I. Schall, Andrew P. Steenhoff, Virginia A. Stallings.

**Writing – review & editing:** Alyssa M. Tindall, Joan I. Schall, Boitshepo Seme, Bakgaki Ratshaa, Michael Tolle, Maria S. Nnyepi, Loeto Mazhani, Richard M. Rutstein, Andrew P. Steenhoff, Virginia A. Stallings.

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
