## [Decision Letter · Decision Letter 0]

30 Mar 2020

PONE-D-20-05581

Vitamin D status, nutrition and growth in HIV-infected mothers and HIV-exposed infants and children in Botswana

PLOS ONE

Dear Dr. Tindall,

Thank you for submitting your manuscript to PLOS ONE. After careful consideration, we feel that it has merit but does not fully meet PLOS ONE’s publication criteria as it currently stands. Therefore, we invite you to submit a revised version of the manuscript that addresses the points raised during the review process.

Thank you for your submission. Please could you respond to the reviewer comments and to those in the attached pdf. 

All reviewers have requested more detail of the methodology and results; the queries are listed in the reviews and pdf.

We would appreciate receiving your revised manuscript by May 14 2020 11:59PM. To enhance the reproducibility of your results, we recommend that if applicable you deposit your laboratory protocols in protocols.io, where a protocol can be assigned its own identifier (DOI) such that it can be cited independently in the future. For instructions see: http://journals.plos.org/plosone/s/submission-guidelines#loc-laboratory-protocols

We look forward to receiving your revised manuscript.

Kind regards,

Emma K. Kalk

Academic Editor

PLOS ONE

Journal Requirements:

Reviewers' comments:

Reviewer's Responses to Questions

**Comments to the Author**

1. Is the manuscript technically sound, and do the data support the conclusions?

Reviewer #1: Yes

Reviewer #2: No

2. Has the statistical analysis been performed appropriately and rigorously? 

Reviewer #1: Yes

Reviewer #2: No

3. Have the authors made all data underlying the findings in their manuscript fully available?

Reviewer #1: Yes

Reviewer #2: No

4. Is the manuscript presented in an intelligible fashion and written in standard English?

Reviewer #1: Yes

Reviewer #2: No

5. Review Comments to the Author

Reviewer #1: The manuscript titled: “Vitamin D status, nutrition and growth in HIV-infected mothers and HIV-exposed infants and children in Botswana.” presented by Tindall et al is reviewed below.

The authors present a paper about vitamin D status and its relation to growth and HIV disease state in women living with HIV and exposed infants. It is a cross-sectional quantitative study. The authors demonstrate that vitamin D insufficiency was relatively rare but that the level of vitamin D did correlate with growth and HIV indicators in mothers.

The information is useful and the work seems to be scientifically sound. The paper is very well written and clear but there are a few issues to address

Methods:

Line 95: Please consider adding details about where mothers were recruited from, was it from their ART follow-up clinics?

Line 111-112: The sentence “the younger infants” seems to conflict with the results in that the sentence says that PCR results were not yet available yet the results clearly distinguish positive and negative. I would say that you cannot present the results as “neg” vs “pos” infants if the negative group actually is largely unknown in terms of HIV status. Please also explain why HIV testing was not offered as part of the study if results were not available on enrolment.

Line 114: The 48 children, were they at all related to the infants and mothers? Were they older siblings or totally unrelated. Please clarify also how and where they were recruited.

Line 132-133: In terms of using both the CDC as well as WHO charts, please explain why and also whether this may cause an effect on the results.

Discussion:

Line 248-249 and also line 264-265: Please consider explaining the relationship between cause and effect of Vit D and various clinical outcomes. Is there evidence that clearly links vitamin D as the cause and height as the effect?

Line 273-274: What comparison are you using the make this statement “Our results also..”

Minor:

Line 45: consider removing one “and” and replace with comma,

Line 86: add the word “vitamin”

Line 262: the word “disease” seems to be missing after HIV.

Line 302: remove “the”

Reviewer #2: The authors are presenting a research study which was designed to determine the vitamin D status in HIV-infected mothers and their children, and to compare vitamin serum levels between HIV positive and negative children.

Major comments

1. The number of participants enrolled is small, only 36 mothers and 48 children. In addition to these small numbers the children of the 36 HIV positive numbers are further divided to HIV-infected to HIV-exposed, with only 9 being HIV infected. This makes it difficult to interpret the significance of findings in this study. How was the sample size calculated, and what were the assumptions made to calculate the sample size?

2. In children, they have stratified them to HIV-positive and HIV-negative, why was the same stratification not done for the mothers?

3. In presenting the Tables, authors have pointed on variables that showed statistical significance using different symbols. It will be much easier for readers to understand the Tables better if they can add a column with p-values.

4. In Table 2, for the differences notes in length and height Z scores, did they adjust for potential confounders e.g. sex, and breast feeding.

5. Table 3 does not reflect on what was being studied, that is vitamin D. Secondly the numbers presented for HIV positive infants and children are confusing. For example for CD4 count there are 2 values, what does this mean. On the row of CD4% the numbers in brackets (8) and (24) are referring to what?

6. In making their conclusion, they need to adjust for potential confounders before they can conclude that children with HIV had poorer linear growth than HIV-uninfected children.

7. The conclusion that there was a significant association between vitamin D status and HIV disease is supported by which results?

8. The authors need to reconsider their statement that the findings from this study show a 'REASSURINGLY LOW PREVALENCE" of vitamin D insufficiency, when they had such a low sample size.

Minor comment

They must try and paraphase some of the sentences in the report as they are often long and one has to read them many times before one can understand their meaning. Below are some of the sentences to be considered for paraphrasing.

1. On page 4, Lines 80-83

2. On page 5, Lines 95-98

3. On page 9, Lines 180-184

6. PLOS authors have the option to publish the peer review history of their article (what does this mean?). If published, this will include your full peer review and any attached files.

Reviewer #1: No

Reviewer #2: No

---

## [Author Response · Author response to Decision Letter 0]

2 Jun 2020

Reviewer #1: The manuscript titled: “Vitamin D status, nutrition and growth in HIV-infected mothers and HIV-exposed infants and children in Botswana.” presented by Tindall et al is reviewed below.

The authors present a paper about vitamin D status and its relation to growth and HIV disease state in women living with HIV and exposed infants. It is a cross-sectional quantitative study. The authors demonstrate that vitamin D insufficiency was relatively rare but that the level of vitamin D did correlate with growth and HIV indicators in mothers.

The information is useful and the work seems to be scientifically sound. The paper is very well written and clear but there are a few issues to address

Methods:

Line 95: Please consider adding details about where mothers were recruited from, was it from their ART follow-up clinics?

Thank you for this suggestion. We have changed the previous sentence to read: Mothers, infants and children were recruited from Princess Marina Hospital, Gaborone, and Bamalete Lutheran Hospital, Ramotswa, in Botswana where they were followed for medical care.

Line 111-112: The sentence “the younger infants” seems to conflict with the results in that the sentence says that PCR results were not yet available yet the results clearly distinguish positive and negative. I would say that you cannot present the results as “neg” vs “pos” infants if the negative group actually is largely unknown in terms of HIV status. Please also explain why HIV testing was not offered as part of the study if results were not available on enrolment.

The HIV PCR testing was provided by the well-established, Botswana Pediatric HIV Program and the study was designed and budgeted for utilizing the results provided through this program. The Botswana 2013 Global AIDS Response Report (when study data were collected) reported that 47% of infants born to HIV positive mothers received a virological HIV test within two months after birth, this is likely why at the time of data collection, the younger infants did not have available results. However, the report also documents that the rate of transmission from mother to child at this time was less than 3%. Given the low transmission rates in this population, these infants were categorized as HIV- at the time of data collection. 

Further, when we removed the infants who did not have a confirmatory test result from the analysis, the significance of the results remained the same for both the analysis between mothers with HIV+ and HIV- infants and between HIV+ and HIV- infants. Specifically, we performed analysis of the data for a subsample of the older infants (6-11.9 months old, n=18) for whom HIV status was known (10 were HIV- and 8 were HIV+). The analyses for this subgroup confirmed our findings presented in the paper: 1) mothers of HIV+ infants had significantly lower 25(OH)D levels and significantly lower CD4% than mothers of HIV- infants; 2) HIV+ infants had poorer growth status than their HIV- counterparts by almost 1 Z score for both length and weight Z scores, although likely due to small sample size this did not reach statistical significance. 

Line 114: The 48 children, were they at all related to the infants and mothers? Were they older siblings or totally unrelated. Please clarify also how and where they were recruited.

The children enrolled in this study were unrelated to the mothers and infants. We have clarified this in the methods section: This was an observational, age- and HIV-status stratified survey of 36 HIV-infected (HIV+) mothers and their infants (0-11.9 months), and 48 unrelated children (1-7.9 years).

Line 132-133: In terms of using both the CDC as well as WHO charts, please explain why and also whether this may cause an effect on the results.

This is the recommendation of the CDC – the CDC recommends using WHO growth charts for children 0-2 years and the CDC growth charts for children 2-18 years. We have amended the sentence in our methods to read: For infants and young children up to age two years where length was measured, length and weight were converted into Z scores (standard deviation units) using the WHO growth chart data (http://www.cdc.gov/growthcharts/), as recommended by the CDC(18). For older children the CDC recommends using the CDC algorithms to convert height and weight into Z scores. Given this is the recommendation for these age groups, we suggest would be inappropriate to use one of the reference charts for all ages.

Discussion:

Line 248-249 and also line 264-265: Please consider explaining the relationship between cause and effect of Vit D and various clinical outcomes. Is there evidence that clearly links vitamin D as the cause and height as the effect?

Thank you for this question. The relation between vitamin D and height is being explored in other clinical research settings. A recently published prospective study examining the effect of micronutrient status and growth in exclusively breastfed children reported: 

“Children who were exclusively breastfed for longer than 4 months without proper supplement were more likely to have persistent vitamin D insufficiency and have relatively slower growth after infancy compared to mix-fed children.” 

Although these children were not exposed to HIV, there was still a relation between vitamin D status and height. The children in this study were Taiwanese and in the discussion of our study, we present studies that report on populations closer to the population we studied to compare vitamin D status and growth, particularly in HIV exposed infants and children.

Line 273-274: What comparison are you using the make this statement “Our results also..”

Thank you for this comment, we have changed the sentence to read: Overall, our results show HIV-exposed infants and children had poorer overall growth regardless of whether they were HIV- or HIV+.

Minor:

Line 45: consider removing one “and” and replace with comma,

We have changed the sentence to read: Length/height and weight Z scores determined growth status.

Line 86: add the word “vitamin”

We have added this.

Line 262: the word “disease” seems to be missing after HIV.

We have added this.

Line 302: remove “the”

We have removed this.

Reviewer #2: The authors are presenting a research study which was designed to determine the vitamin D status in HIV-infected mothers and their children, and to compare vitamin serum levels between HIV positive and negative children.

Major comments

1. The number of participants enrolled is small, only 36 mothers and 48 children. In addition to these small numbers the children of the 36 HIV positive numbers are further divided to HIV-infected to HIV-exposed, with only 9 being HIV infected. This makes it difficult to interpret the significance of findings in this study. How was the sample size calculated, and what were the assumptions made to calculate the sample size?

The primary objective of this study was to determine the vitamin D status of HIV-infected mothers and their infants and children. This was an observational study or survey of vitamin D in Botswana mothers, infants and children. Sample sizes were not calculated based upon effect sizes for the outcomes. Secondary objectives were to examine potential differences in vitamin D status and other outcomes among each of the groups and associations were exploratory. 

2. In children, they have stratified them to HIV-positive and HIV-negative, why was the same stratification not done for the mothers?

Thank you for this question. All of the mothers were HIV+ by study design, so the same stratification is not possible.

3. In presenting the Tables, authors have pointed on variables that showed statistical significance using different symbols. It will be much easier for readers to understand the Tables better if they can add a column with p-values.

We have added the p-values to the table and removed the symbols that described the significance level.

4. In Table 2, for the differences notes in length and height Z scores, did they adjust for potential confounders e.g. sex, and breast feeding.

Thank you for this question. We did examine males and females separately and they had similar growth outcomes. We did adjust for age in the analysis of covariance as this was significant when added to the model. When added to the models, neither sex nor breastfeeding (ever breastfed) were significant covariates, that is, they did not significantly contribute to the growth outcomes and therefore we did not include them in the final model. It is worth noting that very few infants were breastfed (2 in the HIV- group and 1 in the HIV+ group). For the older children, there were no HIV- children who were ever breastfed. A little over half of the HIV+ children had been breastfed. Therefore, we examined using student’s unpaired t tests whether there were differences in growth (length and weight Z scores) among HIV+ children only who were breastfed vs not breastfed and found no significant differences by breastfeeding. Since breastfeeding only occurred in the HIV+ children, we cannot rule out that it may have contributed to the poorer growth status in these children along with other factors, and further studies are be needed to tease this out. 

5. Table 3 does not reflect on what was being studied, that is vitamin D. Secondly the numbers presented for HIV positive infants and children are confusing. For example for CD4 count there are 2 values, what does this mean. On the row of CD4% the numbers in brackets (8) and (24) are referring to what?

Thank you for this comment. Table 3 was included to provide the information on HIV disease status (both immunological and clinical characteristics) for the whole sample. Mothers of HIV exposed (but HIV-) and HIV+ infants could be compared, and mothers of HIV+ infants had poor immunological status (lower CD4%). Therefore, we examined the association in these HIV-infected mothers of CD4% with vitamin D concentration, our primary outcome, and found a significant correlation. 

6. In making their conclusion, they need to adjust for potential confounders before they can conclude that children with HIV had poorer linear growth than HIV-uninfected children.

We appreciate this comment and understand the reviewer’s concern. We have amended the language in the conclusion to address the fact that there may be confounding variables. Breastfeeding may be a confounding variable and a larger sample size and more even distribution amongst children who were breastfed vs formula fed in each group would lend to a better understanding. Additionally, we do not have dietary data to adjust for dietary intake and have listed this as a limitation of this study. We have also suggested future studies should examine confounding factors that could affect this relation as currently, we do not know what other variables may be confounders and this study is not designed for subgroup analyses.

7. The conclusion that there was a significant association between vitamin D status and HIV disease is supported by which results?

This conclusion is from the correlation between the HIV-infected mothers and CD4%: In the HIV-infected mothers as a group, serum 25(OH)D was positively associated with CD4% (r=0.40, p=0.016). 

8. The authors need to reconsider their statement that the findings from this study show a 'REASSURINGLY LOW PREVALENCE" of vitamin D insufficiency, when they had such a low sample size.

We appreciate this suggestion. We have amended the sentence to: Results from this study show a low prevalence of vitamin D insufficiency in this small cohort and emphasize the need for further research to identify confounding variables that affect vitamin D status in HIV infected and exposed individuals.

Minor comment

They must try and paraphase some of the sentences in the report as they are often long and one has to read them many times before one can understand their meaning. Below are some of the sentences to be considered for paraphrasing.

1. On page 4, Lines 80-83

2. On page 5, Lines 95-98

3. On page 9, Lines 180-184

We have restructured these sentences to be shorter and read easier.

---

## [Editor Report · Decision Letter 1]

4 Jun 2020

PONE-D-20-05581R1

Vitamin D status, nutrition and growth in HIV-infected mothers and HIV-exposed infants and children in Botswana

PLOS ONE

Dear Dr. Tindall,

Thank you for submitting your manuscript to PLOS ONE. After careful consideration, we feel that it has merit but does not fully meet PLOS ONE’s publication criteria as it currently stands. Therefore, we invite you to submit a revised version of the manuscript that addresses the points raised during the review process.

Thank you for submitting the revised manuscript. The methodology is clearer and discussion more measured. 

I did not see a response to the notes in the pdf attached to the initial review. 

Below please find further remarks which I hope with strengthen the submission.

We look forward to receiving your revised manuscript.

Kind regards,

Emma K. Kalk

Academic Editor

PLOS ONE

Additional Editor Comments (if provided):

PONE-D-20-05581R1

Line 46: “Immune panel” implies more than CD4 count

Line 43: The groups of infant v child participants have been clarified in the text. Please do so in the abstract.

Line 80: Climate and sunlight are referenced; this could be applied to Botswana. There is already some data on vit D in pregnant women and children and HIV published by your group. you could mention this here.

Line 89: The authors mention nutritional status and feeding in the introduction but seem to have little data on this. Only length/height is discussed and only breast feeding as a binary variable. Duration of breast feeding may also be relevant; age of introduction of other foods, variety of foods etc. It is better not to introduce these terms if you have no related data.

Line 110: Sampling. Please provide more detail on how were these women recruited? Were they attending routine adult HIV services and the asked about off-spring? Were the infants/children recruited first? Was the sample selected randomly? Children >12m living with and without HIV were balanced. How was this done practically, and could it not introduce bias? If the women/children were recruited in a different manner, this could bias the results (we know already they were recruited in different seasons).

Is the hospital site a reflection of the population as a whole? You note that it is an urban population. This limitation should be discussed. As should the omission of all acutely or chronically unwell children who may have a different vitamin D status. It should be clearly stated that these findings are limited to quite a specific group.

Line 113: The amended sentence is not grammatically correct. You could preface this by saying that infants of unconfirmed HIV status were categorized as negative ‘given the rate of perinatal transmission etc.’ Sensitivity analyses are described in the response to the reviewer comment. IT might be useful to mention that sensitivity analyses were performed which did not change the observed associations.

With all the stratification the numbers become very small and once must be careful not to overstate the findings. Caution is required in interpreting these data.

Line 118: adding “unrelated” children would make the groups explicit.

Line 125: HIV status – this is usually positive, negative or indeterminate; could this refer to the clinical and/or immunological stage of women living with HIV?

Line 145: ‘Immunological outcomes’, like ‘an immune panel’ in abstract and ‘immunological markers’ in line is a bit misleading if you mean CD4 and VL only. The terms imply additional assays (cytokines? T cell phenotype?) For example, “Immune status was determined in all women and those children living with HIV…” may be more suitable here.

Line 160: were all data normally distributed?

Line 167: Why did you choose ANCOVA over regression analysis?

Did you check the model?

Please add a description of relevant PMTCT and ART guidelines at the time as these speak to additional exposures.

Line 182: This is a low proportion – is ART not indicated for all pregnant women in Botswana?

Line 187: As presented the sentence states that the HIV infected infants were examined in on season and the HEU in another. Is this correct?

Line 197: breastfeeding. Does this refer to current BF or ever BF?

Line 283: The association was weak; a weak but significant positive correlation. Reviewer 2 had also noted that the original statement is very emphatic.

Are CD4 values presented for the mothers or the infants/children? Conventionally, CD4 is presented as an absolute number in adults (and children >5years) and only as % in the under-fives.

---

## [Author Response · Author response to Decision Letter 1]

9 Jun 2020

Additional Editor Comments:

PONE-D-20-05581R1

Line 46: “Immune panel” implies more than CD4 count

CD4, CD8 and viral load were also measured (HIV-1 RNA VL). We describe the methods in lines 137-145.

Line 43: The groups of infant v child participants have been clarified in the text. Please do so in the abstract.

We have clarified this: This was a cross-sectional study of HIV+ mothers and HIV-exposed infants and unrelated children (1-7.9 years).

Line 80: Climate and sunlight are referenced; this could be applied to Botswana. There is already some data on vit D in pregnant women and children and HIV published by your group. you could mention this here.

We have included an additional citation and sentence here: Vitamin D deficiency is more prevalent in children with perinatally acquired HIV infections(15).

Line 89: The authors mention nutritional status and feeding in the introduction but seem to have little data on this. Only length/height is discussed and only breast feeding as a binary variable. Duration of breast feeding may also be relevant; age of introduction of other foods, variety of foods etc. It is better not to introduce these terms if you have no related data.

We agree, there is a lot of nutritional information that would be helpful in assessing this population. We have amended the sentence to include “breastfeeding” instead of “nutrition”.

Line 110: Sampling. Please provide more detail on how were these women recruited? Were they attending routine adult HIV services and the asked about off-spring? Were the infants/children recruited first? Was the sample selected randomly? Children >12m living with and without HIV were balanced. How was this done practically, and could it not introduce bias? If the women/children were recruited in a different manner, this could bias the results (we know already they were recruited in different seasons).

Is the hospital site a reflection of the population as a whole? You note that it is an urban population. This limitation should be discussed. As should the omission of all acutely or chronically unwell children who may have a different vitamin D status. It should be clearly stated that these findings are limited to quite a specific group.

We have added information to better describe our sample: A convenience sample was used. For balanced representation across infant and child ages and HIV status groups, 18 mother-infant pairs with infants aged 0-5.9 months and 18 pairs with infants aged 6-11.9 months attending routine medical visits were enrolled. (lines 108-110).

We also added the sample as a limitation in the discussion: Additionally, we used a convenience sample that may not be representative of the population the sample size was modest and future larger studies are needed to confirm these data. (line 299)

Line 113: The amended sentence is not grammatically correct. You could preface this by saying that infants of unconfirmed HIV status were categorized as negative ‘given the rate of perinatal transmission etc.’ Sensitivity analyses are described in the response to the reviewer comment. IT might be useful to mention that sensitivity analyses were performed which did not change the observed associations.

With all the stratification the numbers become very small and once must be careful not to overstate the findings. Caution is required in interpreting these data.

Thank you for this comment. We have altered the sentence and added detail about the additional analysis: Given the rate of perinatal transmission of HIV in Botswana was less than 3% at the time of data collection(16), the younger infants (aged 0-5.9 months) were categorized as HIV-. Removing the younger infants did not change the results, therefore, we included them in the dataset.

Line 118: adding “unrelated” children would make the groups explicit.

We have added this.

Line 125: HIV status – this is usually positive, negative or indeterminate; could this refer to the clinical and/or immunological stage of women living with HIV?

We are also classifying how well-controlled/compliant with medication participants are (using CD4 count, viral load).

Line 145: ‘Immunological outcomes’, like ‘an immune panel’ in abstract and ‘immunological markers’ in line is a bit misleading if you mean CD4 and VL only. The terms imply additional assays (cytokines? T cell phenotype?) For example, “Immune status was determined in all women and those children living with HIV…” may be more suitable here.

We have changed the sentence to your recommended language. (line 147)

Line 160: were all data normally distributed?

Variables were assessed for normality using a Skewness and Kurtosis test (Stata) and all primary outcome variables were normally distributed. Some variables were slightly skewed according to the SK test, but quantile plots appeared normal, so variables were not transformed.

Line 167: Why did you choose ANCOVA over regression analysis?

Did you check the model?

We used ANCOVA primarily because we were interested in generating the adjusted means (LS means) for the variables. In the case of the nutritional status variables, such as serum vitamin D, we were interested in the difference between groups (mothers with HIV+ vs. HIV- infants or HIV+ vs. HIV- infants and children) adjusted for the seasonal effect which is known to occur. In the case of the growth variables in the infants and children, we were interested in what the mean differences were in the Z scores between HIV+ and HIV- groups after adjusting for the effect of age. Regression models give much the same results in terms of the significance of the groups differences in these variables, however they do not generate the adjusted means. 

Please add a description of relevant PMTCT and ART guidelines at the time as these speak to additional exposures.

We have added the following language under methods: Between 2006-2009, more than 95% of HIV+ pregnant women in Botswana received the national standard antiretrovirals (ARVs) for prevention of mother to child transmission (PMTCT). This was all HIV infected pregnant women were eligible for triple prophylaxis/ART. HIV infected women were evaluated for ART eligibility based on laboratory testing (CD4 cell count) or clinical presentation (WHO clinical stage). Women with CD4 cell count less than 350 or WHO clinical stage 3 or 4 were started on ART through the national ARV Program as soon as possible, regardless of the stage of pregnancy. Women attending healthcare facilities that provided triple ARV prophylaxis, who were not eligible for ART were provided with triple ARV prophylaxis starting from 14 weeks of gestation. Women who chose to breastfeed continued triple ARV prophylaxis until their infants were at least six months of age and completely after weaning. Both women on ART and those that were on ARV prophylaxis were given supplementary AZT 300 mg every 3 hours during labor and delivery. 

Similarly by 2009, >95% of infants born to HIV infected mothers were given ARV prophylaxis at birth according to national guidelines; they were given Nevirapine as a single dose as soon as possible after birth (within 72 hours) plus four weeks of AZT.

Line 182: This is a low proportion – is ART not indicated for all pregnant women in Botswana?

The paragraphs above explain this – this study was done at the time that ART rollout for all HIV infected pregnant women was being rolled out across Botswana.

Line 187: As presented the sentence states that the HIV infected infants were examined in on season and the HEU in another. Is this correct?

This is correct.

Line 197: breastfeeding. Does this refer to current BF or ever BF?

This refers to “ever” breast fed. We have added “ever” to this sentence.

Line 283: The association was weak; a weak but significant positive correlation. 

Reviewer 2 had also noted that the original statement is very emphatic.

Are CD4 values presented for the mothers or the infants/children? Conventionally, CD4 is presented as an absolute number in adults (and children >5years) and only as % in the under-fives.

We understand the editor and reviewer’s concerns. We have amended the sentence to read: This correlation suggests adequate vitamin D status may be associated with improved HIV disease status.

Additionally, we are only referring to the infected mothers for this correlation: We found a significant, positive correlation between serum 25(OH)D and CD4% in the HIV infected mothers.

---

## [Editor Report · Decision Letter 2]

18 Jun 2020

PONE-D-20-05581R2

Vitamin D status, nutrition and growth in HIV-infected mothers and HIV-exposed infants and children in Botswana

PLOS ONE

Dear Dr. Tindall,

Thank you for submitting your manuscript to PLOS ONE. After careful consideration, we feel that it has merit but does not fully meet PLOS ONE’s publication criteria as it currently stands. Therefore, we invite you to submit a revised version of the manuscript that addresses the points raised during the review process.

I apologize for going backwards and forwards. It is now clear that the women-infant pairs were recruited several years ago (2012 - 2013) and the samples banked. Was this part of an earlier study, perhaps with different objectives? If so, please describe the enrollment procedures of the original study and how the current sample reflects the participant profile of that study. Were the unrelated older children enrolled over the same period in 2012 - 2013? If not, please state this and justify their inclusion. From where were they recruited? As acknowledged by the authors, the sample sizes are small and it is important that there was some system supporting participant inclusion if these results are to be useful. The integrity of the comparator groups needs to be sound and the methodology explicit.

I am still uncertain with respect to the PMTCT guidelines at the time of the study: were these the same as the 2009 guidelines referenced?  

Please include in your abstract and discussion that these findings reflect Vitamin D status in a population prior to the introduction of universal ART in pregnancy.

We look forward to receiving your revised manuscript.

Kind regards,

Emma K. Kalk

Academic Editor

PLOS ONE

---

## [Author Response · Author response to Decision Letter 2]

29 Jun 2020

I apologize for going backwards and forwards. It is now clear that the women-infant pairs were recruited several years ago (2012 - 2013) and the samples banked. Was this part of an earlier study, perhaps with different objectives? If so, please describe the enrollment procedures of the original study and how the current sample reflects the participant profile of that study. Were the unrelated older children enrolled over the same period in 2012 - 2013? If not, please state this and justify their inclusion. From where were they recruited? As acknowledged by the authors, the sample sizes are small and it is important that there was some system supporting participant inclusion if these results are to be useful. The integrity of the comparator groups needs to be sound and the methodology explicit.

Thank you for allowing us to clarify these details. The women-infant pairs and the unrelated children were all specifically recruited for this study which was conducted in 2012-2013. We have also amended the following sentence in the discussion to better characterize the limitations of the study: “we used a convenience sample recruited in 2012-2013 that may not be representative of the population, the sample size was modest and future larger studies are needed to confirm these data.” Although data was collected during this time period, this is the only data that describes vitamin D status and growth in Batswana infants and young children. Hence these data remain important and we hope that the publication of this study will stimulate further work in this area.

I am still uncertain with respect to the PMTCT guidelines at the time of the study: were these the same as the 2009 guidelines referenced? 

Thank you for the opportunity to clarify. The PMTCT guidelines section has been revised to better describe the care guidelines at the time of data collection, which does reflect that ART was available to pregnant women, but implementation was not yet universal.

Please include in your abstract and discussion that these findings reflect Vitamin D status in a population prior to the introduction of universal ART in pregnancy.

We have included this (lines 47-48 & 300-301).

---

## [Editor Report · Decision Letter 3]

9 Jul 2020

Vitamin D status, nutrition and growth in HIV-infected mothers and HIV-exposed infants and children in Botswana

PONE-D-20-05581R3

Dear Dr. Tindall,

We’re pleased to inform you that your manuscript has been judged scientifically suitable for publication and will be formally accepted for publication once it meets all outstanding technical requirements.

Kind regards,

Emma K. Kalk

Academic Editor

PLOS ONE
---

## [Editor Report · Acceptance letter]

3 Aug 2020

PONE-D-20-05581R3 

Vitamin D status, nutrition and growth in HIV-infected mothers and HIV-exposed infants and children in Botswana 

Dear Dr. Tindall:

I'm pleased to inform you that your manuscript has been deemed suitable for publication in PLOS ONE. Congratulations! Your manuscript is now with our production department. 

Kind regards, 

on behalf of

Dr. Emma K. Kalk 

Academic Editor

PLOS ONE